# Evaluation of the Prevalence of Sleep Disorders and Their Association with Stroke: A Hospital-Based Retrospective Study

**DOI:** 10.3390/jcm14041313

**Published:** 2025-02-16

**Authors:** Majed Mohammad Alabdali, Abdulrahim Saleh Alrasheed, Faynan Sultan Alsamih, Reenad Fahad Almohaish, Jumana Nasser Al Hadad, Noor Mohammad AlMohish, Omar Ali AlGhamdi, Suliman Khalid Alabdulaali, Zainab Ibrahim Alabdi

**Affiliations:** 1Neurology Department, College of Medicine, Imam Abdulrahman Bin Faisal University, Khobar 31441, Saudi Arabia; mmalabdali@iau.edu.sa; 2Department of Neurosurgery, College of Medicine, King Faisal University, AlAhsa 31982, Saudi Arabia; 3College of Medicine, King Faisal University, AlAhsa 31982, Saudi Arabia; 220018021@student.kfu.edu.sa (F.S.A.); 219008488@student.kfu.edu.sa (R.F.A.); 4College of Medicine, Jeddah University, Jeddah 23445, Saudi Arabia; 1912893@uj.edu.sa; 5Neurology Department, King Fahad Hospital of the University, Imam Abdulrahman Bin Faisal University, Khobar 31441, Saudi Arabia; nmalmohish@iau.edu.sa (N.M.A.); oaghamdi@iau.edu.sa (O.A.A.); 6Pulmonary and Sleep Medicine Department, King Fahad Hospital, AlAhsa 31982, Saudi Arabia; salabdulaali@moh.gov.sa; 7Neurology Department, King Fahad Hospital, AlAhsa 31982, Saudi Arabia; zalabdi@moh.gov.sa

**Keywords:** stroke, sleep, prevalence, function, rehabilitation, neurological disorders

## Abstract

**Background:** Sleep disturbances are prevalent among stroke survivors, significantly impacting their recovery and quality of life. This study aimed to evaluate the prevalence of sleep disorders, sleep quality, risk of obstructive sleep apnea (OSA), and daytime sleepiness among stroke survivors and to identify potential associations with clinical and demographic factors. **Materials and Methods:** A retrospective observational study analyzed adult stroke survivors (aged ≥ 18 years) attending neurology clinics at our institution from November 2022 to November 2024. The primary outcome measures included overall sleep quality, sleep apnea and daytime sleepiness assessment. Data were collected using validated Arabic versions of the Pittsburgh Sleep Quality Index (PSQI), STOP-Bang Questionnaire, and Epworth Sleepiness Scale (ESS). Statistical analyses, including Chi-square tests and t-tests, were performed using SPSS version 30.1. **Results:** A total of 100 stroke survivors, mostly aged 40–60 years, were recruited in our study. The prevalence of sleep disorders was 60.0%, with poor sleep quality reflected by a mean global PSQI score of 9.13 ± 14.40. Additionally, 19.0% were at high risk of OSA, and 24.0% experienced abnormal daytime sleepiness. While no statistically significant associations were found between sleep disorders and clinical or demographic factors, trends indicated higher sleep disorder prevalence in those with hemorrhagic stroke and high-risk OSA profiles. **Conclusions:** Our study highlights a high prevalence of sleep disorders among stroke survivors, emphasizing the need for regular sleep assessments. Future studies should explore objective assessments and larger sample sizes to validate these findings and to assess their potential implication in stroke recovery and quality of life.

## 1. Introduction

Stroke is the predominant worldwide cause of both morbidity and mortality. It accounts for 11.59% of total global deaths and contributes to 5.65% of the aggregate disability-adjusted life years [1]. It is defined as an acute cerebrovascular disorder characterized by transient or permanent cerebral dysfunction, with stroke categorized into either hemorrhagic or ischemic types based on the underlying pathological mechanisms [2]. Stroke research has increasingly emphasized neuropsychiatric symptoms, reflecting a shift in focus beyond traditional motor outcomes such as the modified Rankin Scale (mRS) and the National Institutes of Health Stroke Scale (NIHSS) [3,4]. Among the neuropsychiatric symptoms, sleep disorders are particularly prevalent, affecting an estimated 25–75% of stroke survivors globally [5,6,7]. These neuropsychiatric symptoms, including sleep disorders, fatigue, and delirium, significantly contribute to the burden of stroke [8,9]. They extend the impact of brain injury beyond physical deficits, influencing emotional well-being, cognitive performance, and functional recovery [10]. Moreover, these symptoms contribute to increased healthcare costs and present substantial challenges to stroke rehabilitation and patient reintegration into society [11,12].

The American Heart Association (AHA) underscores the critical importance of screening for and managing sleep disorders in stroke patients. This guidance emphasizes that identifying and treating sleep disturbances should not be considered secondary but rather integral to comprehensive stroke care [13]. Among these, obstructive sleep apnea (OSA) has garnered particular attention due to its significant impact on stroke outcomes [14]. Early identification and treatment of sleep disorders are now recognized as essential components of comprehensive stroke care, aimed at improving both immediate and long-term recovery [15].

Sleep disorders are not only consequences but also significant risk factors for stroke. They are strongly associated with hypertension, diabetes, obesity, and occupational stress—all established contributors to stroke incidence [5]. Furthermore, untreated sleep disorders increase the likelihood of recurrent stroke and exacerbate stroke-related outcomes, highlighting the bidirectional relationship between sleep disturbances and cerebrovascular health [16].

A wide spectrum of sleep disturbances has been documented in stroke patients. Beyond insomnia and hypersomnia, these include obstructive sleep apnea (OSA), restless leg syndrome (RLS), circadian rhythm sleep disorders, and parasomnias, such as sleepwalking and night terrors. Each of these conditions imposes unique challenges to stroke recovery and necessitates tailored management strategies [16].

Recent theoretical frameworks have shed light on the broader neurobiological implications of sleep in both health and disease. The defensive activation theory posits that sleep serves an adaptive function by preventing excessive cortical excitability and maintaining homeostasis in neural circuits. This theory has been linked to neurological disorders, including stroke, where excessive neuronal activity may contribute to further brain damage and hinder recovery [17]. Active inference theory, on the other hand, suggests that sleep is a crucial process for refining the brain’s predictive models and facilitating optimal sensory-motor integration, thereby enhancing post-stroke rehabilitation outcomes [18]. Emerging research has begun to explore the interplay between these theories and stroke-related sleep disorders, indicating potential therapeutic targets for optimizing recovery through sleep regulation.

The mechanism of sleep is complex and involves multiple neurophysiological and biochemical processes essential for maintaining brain health. Sleep is regulated by the interplay of the circadian rhythm and homeostatic sleep drive, both of which are controlled by the hypothalamus [19]. The suprachiasmatic nucleus (SCN) of the hypothalamus synchronizes sleep–wake cycles based on external light cues, while neurotransmitters, such as gamma-aminobutyric acid (GABA) and orexin, play significant roles in sleep induction and maintenance [20]. Sleep consists of non-rapid eye movement (NREM) and rapid eye movement (REM) stages, each contributing uniquely to cognitive restoration, immune function, and metabolic homeostasis [21].

Good sleep duration and quality are particularly critical for stroke patients, as they facilitate neuroplasticity, enhance synaptic remodeling, and improve cognitive recovery. Sleep plays a vital role in memory consolidation, motor skill learning, and emotional resilience, all of which are necessary for effective stroke rehabilitation. Furthermore, adequate sleep reduces levels of inflammatory cytokines, such as interleukin-6 (IL-6) and tumor necrosis factor-alpha (TNF-α), which have been linked to poor stroke outcomes [22].

Conversely, poor sleep quality and reduced sleep duration can have profound consequences on various aspects of stroke recovery. Disrupted sleep is associated with prolonged hospital stays, increased rates of post-stroke depression, impaired cognitive function, and reduced motor recovery [22]. Additionally, untreated sleep disorders such as OSA increase the risk of stroke recurrence by contributing to intermittent hypoxia, endothelial dysfunction, and sustained hypertension [23]. The cumulative effects of poor sleep on stroke rehabilitation highlight the need for integrating sleep assessments and interventions into standard post-stroke care protocols.

Neuroplasticity is central to post-stroke recovery, and sleep plays a crucial role in supporting this process. Sleep fosters synaptic plasticity, enabling the brain to adapt and repair. It aids in learning, memory consolidation, and motor skill acquisition, which are essential for effective rehabilitation [24]. By improving sleep quality, clinicians can optimize recovery pathways and enhance overall outcomes for stroke patients.

Despite the global focus on post-stroke sleep disorders, to the best of our knowledge, no studies have comprehensively explored their prevalence or impact within Saudi Arabia. Understanding the intricate relationship between sleep and stroke recovery will enable the development of targeted interventions, ultimately improving the quality of life and long-term prognosis for stroke survivors.

## 2. Materials and Methods

### 2.1. Study Design and Population

This study was a retrospective, observational study that analyzed patients data between November 2022 to November 2024. A convenience sampling technique was employed to recruit first-ever adult stroke survivors aged 18 years and older who attended neurology clinics at our institution. The diagnosis of stroke and subtype classification (ischemic vs. hemorrhagic) were confirmed for all participants through radiological imaging (CT and MRI). To ensure the homogeneity of the sample, patients with pre-existing respiratory or cardiac conditions were excluded. Additionally, other medical conditions known to significantly impact sleep quality, including osteoarthritis, polyneuropathies, multiple sclerosis, and neurodegenerative pathologies, were also considered for exclusion. This decision was made to isolate the impact of stroke on sleep disorders and avoid confounding factors that could distort the analysis.

To distinguish new-onset post-stroke sleep disturbances from pre-existing sleep disorders, a thorough review of medical records was performed to identify prior diagnoses of sleep disorders, history of sleep medication use, and lifestyle factors such as nightshift work or frequent intercontinental travel. Patients with documented sleep disorders prior to stroke onset were excluded from the study.

Informed verbal consent was obtained from each participant, and the study was approved by the local Institutional Review Board (IRB) (34A-EP-2024), adhering to ethical guidelines and institutional policies.

### 2.2. Data Collection

Data collection was conducted by four independent senior researchers, who retrieved demographic data (age, gender, marital status, risk factors, and geographic location), clinical characteristics (stroke subtype, admission date, and stroke location), and neuropsychiatric symptoms from electronic records. Data on sleep difficulties experienced by stroke survivors were collected using a structured pre-piloted Excel sheet (Redmond, WA, USA) documenting sleep variables from patient records and standardized questionnaires.

The assessment of post-stroke sleep disturbances was conducted at one time point after discharge to home, and these assessments aimed to minimize potential hospital-related confounders such as noise, stress, and disruption of sleep routines.


*
**Data Collection Tools:**
*


**The Pittsburgh Sleep Quality Index (PSQI):** A validated 19-item self-report questionnaire was used to assess overall sleep quality and disturbances over the past month. It consisted of seven component scores, including subjective sleep quality, sleep latency (the time taken to fall asleep), sleep duration, habitual sleep efficiency (the percentage of time spent asleep while in bed), sleep disturbances, use of sleeping medication, and daytime dysfunction. Each component was scored on a scale from 0 to 3, and the total PSQI score was obtained by summing the seven components, with scores ranging from 0 to 21. A higher score indicates poorer sleep quality. A PSQI score greater than 5 is commonly used as a threshold to define poor sleep quality, making this tool useful for distinguishing between individuals with good and poor sleep patterns. The PSQI has been widely validated in different populations, including those with neurological conditions, making it particularly relevant for stroke survivors [25].

**The Epworth Sleepiness Scale (ESS):** An eight-item questionnaire was designed to measure excessive daytime sleepiness by evaluating the likelihood of dozing off in various common daily situations, such as sitting and reading, watching television, sitting in a public place, being a passenger in a car, lying down to rest, sitting and talking to someone, sitting quietly after lunch, and driving in traffic. Each situation was rated on a scale of 0 (no chance of dozing) to 3 (high chance of dozing), with a total possible score ranging from 0 to 24. Higher scores indicate greater levels of daytime sleepiness, with scores above 10 suggesting excessive sleepiness requiring further assessment. The ESS is particularly useful for identifying individuals at risk of sleep disorders, including obstructive sleep apnea (OSA), narcolepsy, and other hypersomnolence conditions, and is commonly used in both clinical and research settings [26].

**The STOP-Bang Questionnaire:** An eight-item screening tool specifically designed for the identification of obstructive sleep apnea (OSA) was used. The acronym STOP-Bang stands for Snoring, Tiredness, Observed apneas, high blood Pressure, Body Mass Index (BMI) > 35 kg/m^2^, Age > 50 years, Neck circumference > 40 cm, and Gender (male). Each item was scored as either 0 (no) or 1 (yes), with a total score ranging from 0 to 8. A score of 3 or more was indicative of a high risk for OSA, with higher scores correlating with increased severity of the condition. The STOP-Bang Questionnaire is widely used due to its high sensitivity in detecting moderate-to-severe OSA, making it an effective tool for screening in clinical populations, including stroke survivors who may have an increased prevalence of sleep-disordered breathing [27].

### 2.3. Outcome Measures and Statistical Analysis

The primary outcome measures included overall sleep quality, sleep apnea risk, and daytime sleepiness. Secondary outcome measures included associations with demographic and clinical factors. Sleep disorders were categorized separately from other potential contributors such as OSA to avoid misinterpretation.

Data analysis was conducted using the Statistical Package for the Social Sciences (SPSS), version 30.1 (Chicago, IL, USA). Qualitative variables were analyzed using the Chi-square test to establish associations. Sleep disorder scores were summarized using either mean and standard deviation or median and interquartile range, depending on the distribution. Independent *t*-tests were used to compare sleep-related measures across different stroke subtypes and patient characteristics, with normality assessed using the Shapiro–Wilk test. For non-normally distributed data, the Mann–Whitney U test was applied.

This methodological approach ensures that only post-stroke sleep disorders were evaluated, minimizing hospital-related confounders, and distinguishing sleep disorders from other related conditions such as OSA, thereby providing a clearer understanding of the impact of stroke on sleep patterns.

## 3. Results

### 3.1. Participants Characteristics

The study included 100 patients, with a majority being male (60.0%) and Saudi nationals (99.0%). Regarding age distribution, most patients were aged between 51 and 59 years (28.0%), followed by those aged 40 to 50 years (24.0%). The predominant stroke type was cerebral infarction, accounting for 76.0%, while hemorrhagic strokes were less common (13.0%). Regarding comorbidities, 32.0% of the participants reported having both hypertension (HTN) and diabetes mellitus (DM), while 29.0% reported no associated diseases (Table 1).

### 3.2. Overall Sleep Quality

The mean global Pittsburgh Sleep Quality Index (PSQI) score was 9.13 with a standard deviation of 14.40, indicating overall poor sleep quality among the participants. Among the seven PSQI components, the highest scores were observed in sleep latency (mean = 1.59, SD = 1.18) and sleep disturbance (mean = 1.18, SD = 0.61). The lowest mean score was for sleep efficiency (mean = 0.24, SD = 0.64), reflecting variability in different aspects of sleep quality among the stroke survivors (Table 2).

### 3.3. Sleep Apnea

Sleep disorders were prevalent in 60.0% of the participants. The assessment of obstructive sleep apnea (OSA) risk using the STOP-Bang Questionnaire revealed that 19.0% of the participants were at high risk, while 32.0% were at intermediate risk. Approximately half of the participants (49.0%) were classified as having a low risk of OSA (Table 3).

### 3.4. Daytime Sleepiness

Based on the Epworth Sleepiness Scale (ESS), 24.0% of the participants exhibited abnormal daytime sleepiness, while 76.0% had normal levels. This suggests that a significant proportion of stroke survivors in the study experienced excessive daytime sleepiness (EDS) (Table 3).

### 3.5. Sleep Disturbance and Associated Factors

The prevalence of sleep disorders varied across clinical and demographic factors, though no statistically significant associations were observed. Among stroke types, participants with hemorrhagic stroke showed a prevalence of sleep disorders of 69.3%, while for participants with ischemic cerebral infarction, the rate was 60.5%. However, the difference could not be corroborated statistically. For the comorbidities, no modifying effect of HTN, DM, their combination, or sickle cell anemia on the rate of sleep disturbance was documented.

Sleep disorders were more common in males (64.5%) than females (52.6%), although the difference was not significant (*p* = 0.244). Similarly, participants classified as high-risk for OSA showed a higher prevalence of sleep disorders (73.7%) compared to intermediate (62.5%) and low-risk groups (53.1%), though the association was not statistically significant (*p* = 0.280). Those with abnormal daytime sleepiness had a higher prevalence of sleep disorders (70.8%) compared to those with normal daytime sleepiness (56.6%), with no significant association (*p* = 0.214) (Table 4).

## 4. Discussion

### 4.1. Summary of Results

The present study investigated the clinical and demographic factors associated with sleep disturbances among stroke survivors, assessing their sleep quality, risk of obstructive sleep apnea (OSA), and daytime sleepiness. The findings reveal a significant prevalence of post-stroke sleep disorders, with 60.0% of participants experiencing some form of sleep disturbance. The mean global Pittsburgh Sleep Quality Index (PSQI) score of 9.13 (SD = 14.40) reflects overall poor sleep quality in this population.

The risk of OSA, assessed using the STOP-Bang Questionnaire, revealed that 19.0% of participants were at high risk, 32.0% at intermediate risk, and 49.0% at low risk. Furthermore, the Epworth Sleepiness Scale (ESS) indicated that 24.0% of participants exhibited abnormal daytime sleepiness, reflecting a notable proportion of stroke survivors struggling with excessive daytime sleepiness.

The analysis of clinical and demographic factors showed variations in the prevalence of sleep disorders. Sleep disturbance was insignificantly higher among male, hemorrhagic stroke patients, patients with hypertension and diabetes mellitus, and in those with abnormal daytime sleepiness. Although no statistically significant associations were identified across the analyzed factors, the findings highlight that the prevalence of post-stroke sleep disorders remains high, suggesting that these disturbances are a widespread issue among stroke survivors. The observed patterns, particularly the high prevalence of poor sleep quality, elevated OSA risk, and excessive daytime sleepiness may underscore the critical need for targeted interventions to address such issues in stroke survivors.

### 4.2. Hemorrhagic Stroke and Poorer Sleep Quality

The present study found that sleep disturbances were insignificantly higher among hemorrhagic stroke patients compared to ischemic stroke patients, a finding that aligns with previous literature [8,28] but did not reach statistical significance in our sample. This suggests that while hemorrhagic stroke patients may experience worse sleep quality, other unaccounted factors, such as sample size, lesion variability, and comorbid conditions, may influence these findings [22]. Previous studies have consistently reported that hemorrhagic stroke is associated with greater sleep disruption due to its profound effects on brain structures involved in sleep regulation, particularly the brainstem and thalamus [29,30]. The extensive neuronal damage and inflammatory response triggered by hemorrhagic stroke can impair sleep–wake cycles, disrupt circadian rhythm regulation, and lead to increased sleep fragmentation [31].

One possible explanation for our findings is that hemorrhagic stroke patients in our study had a wide range of lesion locations, some of which may not have directly affected sleep-regulating structures [28]. The heterogeneity of lesion sites across patients could have diluted the expected association between hemorrhagic stroke and more severe sleep disturbances [8]. Additionally, the use of subjective sleep assessment tools, such as the PSQI, STOP-Bang Questionnaire, and ESS, may not have captured the full extent of sleep disorders, especially central sleep apnea (CSA), which is prevalent in hemorrhagic stroke but not effectively screened by STOP-Bang [31]. Given that CSA results from autonomic dysfunction caused by damage to brainstem respiratory centers, the lack of objective measures such as polysomnography in our study may have underestimated its true prevalence.

Another possible mechanism for our findings is the role of intracranial pressure and cerebral edema, which are more pronounced in hemorrhagic strokes and can disrupt sleep due to increased discomfort, headaches, and autonomic instability [28]. However, it is possible that medical management strategies, such as early interventions to control intracranial pressure and blood pressure, may have mitigated some of the expected sleep disturbances in our cohort. Furthermore, psychological factors, including anxiety and depression, which are known to be more prevalent in hemorrhagic stroke patients, may have contributed to poor sleep but were not directly analyzed in our study [28]. The subjective nature of sleep assessments may also have influenced how patients perceived their sleep quality, potentially masking significant physiological sleep disturbances.

In contrast, some studies have found an even stronger association between hemorrhagic stroke and severe sleep disorders, particularly in cases with brainstem involvement [22]. This highlights the importance of considering lesion location when interpreting sleep outcomes.

### 4.3. Sleep Processes and Cortical Areas Involved

Sleep plays a vital role in neurological recovery, particularly in processes involving the prefrontal cortex and hippocampus, which are critical for memory and learning [32]. The prefrontal cortex governs higher-order cognitive functions, including decision-making, problem-solving, and executive control, while the hippocampus is central to memory consolidation, emotional regulation, and spatial navigation [33]. Disruption of these areas due to poor sleep impairs cognitive recovery, emotional stability, and overall rehabilitation outcomes in stroke survivors [34]. Our findings align with existing literature demonstrating that stroke survivors frequently experience sleep fragmentation, increased nocturnal awakenings, and reduced rapid eye movement (REM) sleep, all of which can negatively affect cognitive function [35].

### 4.4. Effects of Sleep on Stroke Recovery

Proper sleep is indispensable for stroke recovery, influencing multiple dimensions of rehabilitation and recovery. Adequate sleep is a cornerstone in the cognitive recovery process, significantly enhancing memory, attention, executive function, and problem-solving abilities [35,36]. Inadequate sleep, on the other hand, can impair memory consolidation and hinder the ability to adapt to rehabilitation programs, delaying recovery and possibly contributing to cognitive decline [37].

Quality sleep helps mitigate common post-stroke emotional challenges, such as depression, anxiety, irritability, and emotional lability, fostering greater emotional stability and mental resilience during the recovery journey [32,38]. Stroke survivors with poor sleep quality may experience worsened psychological symptoms, such as low mood and anxiety, which can further complicate their rehabilitation efforts and hinder their recovery [33]. Moreover, sufficient sleep is essential for social reintegration, as it enhances interpersonal relationships, communication skills, and overall social functionality. Stroke survivors with poor sleep may struggle with social isolation, leading to a decline in their quality of life and making it more challenging to reintegrate into their communities [38,39].

Beyond cognitive, emotional, and social dimensions, sleep is also integral to physical recovery, promoting tissue repair, reducing inflammation, and regulating hormones necessary for healing. Studies have shown that poor sleep can lead to increased markers of inflammation, which can adversely affect the recovery process by hindering tissue repair and prolonging the healing of damaged areas [39].

### 4.5. Physician Management and Guidelines

Currently, physician management of post-stroke sleep disturbances often includes tools such as the PSQI, STOP-Bang Questionnaire, and ESS for sleep disorders screening [8,40]. However, specific guidelines for managing sleep disorders in stroke patients remain limited. The American Heart Association (AHA) emphasizes the importance of screening for conditions like OSA and suggests early interventions such as continuous positive airway pressure (CPAP) therapy for high-risk individuals [41]. Incorporating cognitive behavioral therapy for insomnia (CBT-I) and other non-pharmacological approaches can further enhance recovery outcomes [42]. This study underscores the need for comprehensive guidelines addressing the identification and management of sleep disorders in stroke care.

### 4.6. Molecular and Biological Basis of Sleep Disorders in Post-Stroke Patients

The molecular basis of post-stroke sleep disturbances involves dysregulation of neurotransmitters such as serotonin, dopamine, and gamma-aminobutyric acid (GABA), which are integral to sleep–wake cycles [43,44]. Additionally, inflammation and oxidative stress following a stroke can disrupt normal brain function, contributing to sleep disturbances [8]. Understanding these mechanisms can guide targeted therapies aimed at restoring sleep architecture, thereby improving clinical outcomes and facilitating neuroplasticity in recovery.

### 4.7. Community Burden and Economic Costs

The burden of sleep disorders in stroke patients extends beyond individual health, imposing substantial economic and social costs on communities and healthcare systems. Prolonged recovery times often lead to extended hospital stays and increased outpatient visits, significantly elevating healthcare expenditures [45]. For instance, untreated sleep disorders such as obstructive sleep apnea (OSA) can necessitate frequent readmissions and additional medical interventions, further straining healthcare resources. These conditions also contribute to reduced productivity, as stroke survivors may face challenges reintegrating into the workforce due to persistent fatigue, as well as cognitive impairment and emotional instability. At a societal level, the loss of workforce participation and caregiver absenteeism compounds the economic impact. Moreover, the applications of those low-cost questionnaires to patients before discharge and during follow-ups could be useful in the early diagnosis of sleep disorders using inpatient screening studies, reducing the gap between the onset of the condition, diagnosis, and treatment. Addressing sleep disturbances early in stroke patients can mitigate these costs by accelerating recovery, reducing dependency on healthcare services, and improving overall quality of life, ultimately alleviating the economic burden on families and communities [46].

### 4.8. Comparison with Relevant National and Local Studies

The prevalence of sleep disorders in this study was 60.0%, consistent with other research reporting sleep disorders in 50–75% of stroke survivors [8]. In addition, similar to our results, a previous study conducted in Egypt showed that 70.6% of patients with stroke had sleep disorders, while 61.6% and 20% had poor sleep quality and a severe degree of excessive daytime sleepiness [47]. Moreover, another study reported a prevalence of self-reported post-stroke sleep disturbances (PSSD) of 72% using PSQI with high risk of OSA of 33% [7]. In addition, a recent meta-analysis reported that the prevalence of insomnia or insomnia symptoms after stroke was 38.2% (95% CI: 30.1–46.5) [48]. Sleep-disordered breathing, including OSA, is particularly common in this population and is a known risk factor for recurrent strokes and poor recovery outcomes [49].

In addition, another study conducted among 437 patients with ischemic stroke reported a prevalence of EDS and insomnia of 10–14% and 20–28% [47]. Moreover, another cross-sectional study reported that out of 112 patients with previous stroke, 52.7% reported fatigue and 64.3% reported poor sleep quality [50]. In the present study, 19.0% of participants were classified as high-risk for OSA based on the STOP-Bang Questionnaire, while 32.0% were at intermediate risk. These results highlight the importance of routine screening for OSA among stroke survivors to identify high-risk individuals and implement early treatment, such as continuous positive airway pressure (CPAP) therapy, which has shown benefits in improving outcomes [37].

### 4.9. Strengths and Limitations

Despite the growing recognition of the impact of sleep disorders on stroke recovery, there is a notable lack of data on post-stroke sleep disorders (PSSD) in Saudi Arabia. This gap in knowledge underscores the urgency of conducting further research to better understand the prevalence, types, and implications of sleep disturbances in this population. Establishing evidence-based guidelines to assess and manage sleep disorders among stroke survivors is critical, as these conditions significantly influence rehabilitation outcomes. Such guidelines could enhance post-stroke care by addressing sleep disturbances early, potentially improving neuroplasticity, functional recovery, and overall quality of life. This need extends beyond Saudi Arabia, emphasizing the global importance of integrating sleep assessments into stroke rehabilitation protocols to optimize recovery outcomes worldwide. A significant limitation of the current study is the lack of information about important factors such as smoking that are known factors affecting the incidence of sleep disorders. In future studies, those factors should be assessed in order to show whether they could interfere with the current results. Moreover, all measures used to assess sleep disorders in the current study are subjective while there are no objective data such as polysomnography or actigraphy to confirm the diagnosis of sleep disturbance. Future research incorporating neuroimaging data, objective sleep assessments such as polysomnography, and larger sample sizes may provide a clearer understanding of the relationship between stroke and sleep disturbances. Our findings, while in line with existing literature, suggest that more comprehensive screening strategies are needed to accurately assess sleep disorders in stroke patients to ensure timely diagnosis and potential intervention.

### 4.10. Future Directions

Future research should focus on improving screening processes for sleep disorders, integrating objective assessments such as polysomnography, and exploring the molecular mechanisms underlying sleep disturbances in stroke patients. Additionally, developing comprehensive guidelines and intervention strategies for sleep management in stroke rehabilitation could enhance patient outcomes. Expanding the scope of studies to include diverse populations and evaluating the long-term impact of targeted interventions will further refine clinical practice and contribute to better post-stroke care.

## 5. Conclusions

In conclusion, this study underscores the significant burden of sleep disorders among stroke survivors, revealing that sleep disturbances are a prevalent and critical issue in this population. The high rates of poor sleep quality, increased risk of obstructive sleep apnea (OSA), and excessive daytime sleepiness observed among participants emphasize the importance of addressing sleep-related issues in post-stroke rehabilitation.

Additionally, while no statistically significant associations were found between specific clinical and demographic factors and sleep disturbances, the observed findings highlight potential areas for further investigation. Factors such as stroke type (hemorrhagic vs. ischemic), comorbid conditions such as hypertension and diabetes, and gender may influence the prevalence and severity of sleep disorders and could warrant more targeted research in future studies. The findings also stress the need for individualized and comprehensive screening protocols that consider the various factors contributing to sleep disturbances in stroke survivors.

## Figures and Tables

**Table 1 jcm-14-01313-t001:** Clinical and demographic factors of the patients.

		Number	Percentage
Stroke type	Cerebral infarction	76	76.0%
	Hemorrhagic	13	13.0%
	Not specified stroke	11	11.0%
What diseases do you suffer from?	Nothing	29	29.0%
	HTN	27	27.0%
	DM	10	10.0%
	HTN and DM	32	32.0%
	Sickle cell anemia	2	2.0%
Gender	Male	60	60.0%
Age	Younger adult	4	4.0%
	Adults	36	36.0%
	Older adults	60	60.0%
Nationality	Saudi	99	99.0%

HTN: Hypertension; DM: Diabetes mellitus.

**Table 2 jcm-14-01313-t002:** The Pittsburgh Sleep Quality Index (PSQI) components and global scores.

	Mean	Standard Deviation
Component 1: Subjective sleep quality	0.86	0.93
Component 2: Sleep latency	1.59	1.18
Component 3: Sleep duration	0.94	0.93
Component 4: Sleep efficiency	0.24	0.64
Component 5: Sleep disturbance	1.18	0.61
Component 6: Use of sleep medication	0.33	0.91
Component 7: Daytime dysfunction	1.04	0.98
Global PSQI Score	9.13	14.40

**Table 3 jcm-14-01313-t003:** Post-stroke sleep disorders using simply administered questionnaires.

		Number	Percentage
Pittsburg Sleep Quality Index (PSQI)	Equal or less than 5	40	40.0%
	More than 5	60	60.0%
Epworth Sleepiness Scale (ESS)	Equal or less than 16	76	76.0%
	More than 16	24	24.0%
OSA risk using STOP-Bang Questionnaire	OSA-Low risk	49	49.0%
	OSA-Intermediate risk	32	32.0%
	OSA-High risk	19	19.0%

OSA: Obstructive sleep apnea.

**Table 4 jcm-14-01313-t004:** The association between prevalence of sleep disorders and other clinical and demographic factors of the patients.

		Sleep Disorder				
		No		Yes		*p*-Value
		Number	Percentage	Number	Percentage	
Stroke type	Cerebral infarction	30	39.5%	46	60.5%	0.503
	Hemorrhagic	4	30.7%	9	69.3%	
	Not specified stroke	6	54.5%	5	45.5%	
What diseases do you suffer from?	Nothing	11	37.9%	18	62.1%	0.356
	HTN	14	53.8%	12	46.2%	
	DM	2	20.0%	8	80.0%	
	HTN and DM	12	36.4%	21	63.6%	
	Sickle cell anemia	1	50.0%	1	50.0%	
Gender	Male	22	35.5%	40	64.5%	0.244
	Female	18	47.4%	20	52.6%	
Age	Younger adults	1	25.0%	3	75.0%	0.802
	Adults	17	48.5%	18	51.5%	
	Older adults	22	36.06%	39	63.94%	
STOP-Bang Questionnaire	OSA—Low Risk	23	46.9%	26	53.1%	0.280
	OSA—Intermediate Risk	12	37.5%	20	62.5%	
	OSA—High Risk	5	26.3%	14	73.7%	
Epworth Sleepiness Scale (ESS)	Normal	33	43.4%	43	56.6%	0.214
	Abnormal daytime sleepiness	7	29.2%	17	70.8%	

HTN: Hypertension; DM: Diabetes mellitus OSA: Obstructive sleep apnea.

## Data Availability

Data are available upon reasonable request from the corresponding author.

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
