# Peer review of "Evaluation of the Prevalence of Sleep Disorders and Their Association with Stroke: A Hospital-Based Retrospective Study"

_jcm, 2025, doi:10.3390/jcm14041313_

Round 1

Reviewer 1 Report

Comments and Suggestions for Authors

"Evaluation of the Prevalence and Functional Impact of Post-Stroke Sleep Disorders: A Hospital-Based Retrospective Study" is a very important and interesting study and should be published.

The methodology has to be extended (especially description of questionnaires).

The tables e.g. percentage aren't clear enough (in comparison to total cases). 

Comments on the Quality of English Language

The linguistic style could be improved and should be cheched by a native speaker.

Author Response

Comment 1:
[The methodology has to be extended (especially description of questionnaires).].

Reply 1:
[Dear Reviewer, thank you for your insightful feedback. Please find the added comments on pages 3-5, lines 141-219].

Comment 2: 
[The tables e.g. percentage aren't clear enough (in comparison to total cases). ].

Response 2: 
[Dear Reviewer, thank you for your insightful feedback. Please be advised that the percentages are calculated either based on the total cases (i.e., 100) or the subtotal according to the table and the parameters used].

Comment 3:

[The linguistic style could be improved and should be cheched by a native speaker.].

Response 3: 
[Dear Reviewer, thank you for your insightful feedback. Please be advised that the language has been reviewed and refined by experts in the field.].

Reviewer 2 Report

Comments and Suggestions for Authors

Title: Evaluation of the Prevalence and Functional Impact of Post-Stroke Sleep Disorders: A Hospital-Based Retrospective Study

Brief Summary: this is a retrospective study of stroke survivors at a single center within defined time frame. Subjective questionnaires utilized to study 100 patients to evaluate prevalence of sleep disturbances in this population. PSQI, STOP-Bang and Epworth sleepiness scale were utilized.

60% of patients noted to have poor sleep quality based on the PSQI scores. No association found between presence of sleep disturbances and clinical or demographic variables. There was a positive signal with higher prevalence of sleep disorders in patient with hemorrhagic stroke and higher scoring patients on STOP Bang questionnaire.

General comments:

1.       Ideally, authors should consider screening all 100 patients studied using outpatient sleep study and correlate their results.

2.       Recommend discussing potential use of inpatient sleep screens for patients with identified higher risk based on questionnaires

3.       Hemorrhagic stroke patients would have higher prevalence of central sleep apnea which is not likely to be flagged by STOP-Bang in the immediate recovery period. Also STOP-Bang is notoriously underperformed in women as per literature.

4.       Authors should elaborate in discussion on how to incorporate low cost measures for these patients upon discharge to reduce bridging time to diagnosis and treatment.

Specific comments:

1.       Table 1 – only need to indicate one gender, second is assumed as 100-given%.

2.       Table 1 – age groups can be condensed into three groups, young adult, adult and older adults

Author Response

Comment 1: 
[Ideally, authors should consider screening all 100 patients studied using outpatient sleep study and correlate their results.].

Response 1: 
[Dear Reviewer, thank you for your insightful feedback. We appreciate your valuable input. Please note that we attempted to conduct an objective intervention; however, due to certain limitations, we were unable to do so. We have acknowledged this as a limitation on page 10, paragraph 4, lines 314–317.].

Comment 2:
[Recommend discussing potential use of inpatient sleep screens for patients with identified higher risk based on questionnaires].

Response 2:
[Dear Reviewer, thank you for your insightful feedback. Please find the added comments on page 13, lines 443-446].

Comment 3:

[Hemorrhagic stroke patients would have higher prevalence of central sleep apnea which is not likely to be flagged by STOP-Bang in the immediate recovery period. Also STOP-Bang is notoriously underperformed in women as per literature.].

Response 3: 
[Dear Reviewer, thank you for your insightful feedback. Please find the changes on pages 10-11, lines 295-330].

Comment 4: 
[  Authors should elaborate in discussion on how to incorporate low cost measures for these patients upon discharge to reduce bridging time to diagnosis and treatment.].

Response 4: 
[Dear Reviewer, thank you for your insightful feedback. Please find the added comments on page 13, lines 443-446].

Comment 5: 
[Table 1 – only need to indicate one gender, second is assumed as 100-given%.].

Response 5:
[Dear Reviewer, thank you for your insightful feedback. Please find the changes on page 6 in Table 1.].

Comment 6:
[Table 1 – age groups can be condensed into three groups, young adult, adult and older adults].

Response 6: 
[Dear Reviewer, thank you for your insightful feedback. Please find the changes on page 6 in Table 1.].

Reviewer 3 Report

Comments and Suggestions for Authors

The paper by Majed Mohammad Alabdali et al. is a retrospective study assessing the prevalence of sleep disorders in stroke patients in Saudi Arabia. Although the topic is certainly of interest given the implications in terms of neurorehabilitation quality of life, the study does not add much to what is already known in the literature on the link between stroke and stroke except that it explores the characteristics of Saudi Arabia, which is less represented than other regions of the world in the literature in this context. In addition to this, numerous aspects require, in my opinion, extensive review: 

- In the introduction, I suggest providing some conceptual detail on the role of sleep, perhaps in light of some recent interpretations. Indeed, recent papers may provide a neurobiological rationale for the correlation between sleep and measures of interest to be further explored in the Discussion. For example, recent studies have evaluated correlations between the Defensive Activation Theory, Active Inference Theory, and various neurological and psychiatric disorders, e.g. positive neurological symptoms, but also in healthy populations. I think these additions could help better understand the results' implications and suggest new insights into their interpretation from a pathophysiological and functional point of view.  

- In the Methods, it is unclear to me why only cardio-respiratory problems and stroke recurrence are used as exclusion criteria; numerous conditions may alter the sleep profile and that were not considered in the data collection (e.g. osteoarthritis, polyneuropathies, multiple sclerosis, neurodegenerative pathology, to name a few). The Authors should justify this choice; 

- Furthermore, among the comorbidities of interest, smoking or other factors that may be both risk factors for stroke and sleep disorders were not evaluated; these data should be supplemented and their implications assessed;

- Much more critical is the fact that it is not explained when and for how many times the outcome measures of interest on sleep were assessed (at the beginning of hospitalisation? during hospitalisation? at discharge? other?), which is crucial because hospitalisation may alter the sleep profile (noisy ward, concern about the health condition, etc.) but these issues may also disappear on return home; providing this information will be crucial; 

- Another essential aspect is that it is not clear whether the patients included were already suffering from sleep disturbances before the stroke (which, as the Authors rightly note, is itself a risk factor for stroke). For example, it is not assessed whether they took sleeping medication or whether they worked night shifts/frequent intercontinental travel (for example). This is crucial to distinguish actual post-stroke sleep disorders from pre-existing problems; 

- Another critical aspect is not to confuse sleep disorders per se with problems that may arise from other disorders and influence the sleep pattern (e.g., OSA); I, therefore, suggest keeping these concepts separate; 

- The Discussion has little focus on the results and explores more general topics of context, whereas it should discuss more the main findings in the current literature, e.g., why do haemorrhagics seem to sleep worse? Are lesional areas in the recruited sample that seem to be associated with more significant sleep disturbances? For what reasons?

- In the title, the Authors state that they also evaluate the functional impact of these issues, but no data (e.g., Barthel's index, mRS, IADL) is reported, which would suggest what the association is with post-stroke functional status; I suggest that these data be included or the text amended accordingly; 

- In this work (doi: 10.1007/s10072-024-07815-y), the Authors did not observe a correlation between sleep measures (some of which were also used in this study) and motor learning performance; I suggest citing this and other work in the discussion to explore differences from other literature data. For example, their stroke timeframe was subacute/chronic, whereas, here, the Authors assessed earlier stages. What are the potential implications? 

- Amongst the limitations, I also suggest reporting that all measures used are subjective, while there are no objective data (e.g., polysomnography, actigraphs) to confirm the actual presence of sleep disturbances. 

Author Response

Comment 1:

[In the introduction, I suggest providing some conceptual detail on the role of sleep, perhaps in light of some recent interpretations. Indeed, recent papers may provide a neurobiological rationale for the correlation between sleep and measures of interest to be further explored in the Discussion. For example, recent studies have evaluated correlations between the Defensive Activation Theory, Active Inference Theory, and various neurological and psychiatric disorders, e.g. positive neurological symptoms, but also in healthy populations. I think these additions could help better understand the results' implications and suggest new insights into their interpretation from a pathophysiological and functional point of view.]

Response 1: 

Dear Reviewer, thank you for your insightful feedback. Please find the added comments on pages 2-3, lines 83-128.

Comment 2:

[In the Methods, it is unclear to me why only cardio-respiratory problems and stroke recurrence are used as exclusion criteria; numerous conditions may alter the sleep profile and that were not considered in the data collection (e.g. osteoarthritis, polyneuropathies, multiple sclerosis, neurodegenerative pathology, to name a few). The Authors should justify this choice].

Response 2:

[Dear Reviewer, thank you for your insightful feedback. Please find the added comments on page 4, lines 140-149].

Comment 3:

[Furthermore, among the comorbidities of interest, smoking or other factors that may be both risk factors for stroke and sleep disorders were not evaluated; these data should be supplemented and their implications assessed].

Response 3:

[Dear Reviewer, thank you for your insightful feedback. Please find the added comments on page 14, lines 481-483].

Comment 4:

[Much more critical is the fact that it is not explained when and for how many times the outcome measures of interest on sleep were assessed (at the beginning of hospitalisation? during hospitalisation? at discharge? other?), which is crucial because hospitalisation may alter the sleep profile (noisy ward, concern about the health condition, etc.) but these issues may also disappear on return home; providing this information will be crucial].

Response 4:

[Dear Reviewer, thank you for your insightful feedback. Please find the added comments on page 4, lines 167-169].

Comment 5:

[Another essential aspect is that it is not clear whether the patients included were already suffering from sleep disturbances before the stroke (which, as the Authors rightly note, is itself a risk factor for stroke). For example, it is not assessed whether they took sleeping medication or whether they worked night shifts/frequent intercontinental travel (for example). This is crucial to distinguish actual post-stroke sleep disorders from pre-existing problems].

Response 5:

[Dear Reviewer, thank you for your insightful feedback. Please find the added comments on page 4, lines 152-156].

Comment 6:

[Another critical aspect is not to confuse sleep disorders per se with problems that may arise from other disorders and influence the sleep pattern (e.g., OSA); I, therefore, suggest keeping these concepts separate].

Response 6:

[Dear Reviewer, thank you for your insightful feedback. Please find the added comments on page 7, lines 239-245].

Comment 7:

[The Discussion has little focus on the results and explores more general topics of context, whereas it should discuss more the main findings in the current literature, e.g., why do haemorrhagics seem to sleep worse? Are lesional areas in the recruited sample that seem to be associated with more significant sleep disturbances? For what reasons?].

Response 7: 
[Dear Reviewer, thank you for your insightful feedback. Please find the changes on pages 10-11, lines 295-330].

Comment 8:

[In the title, the Authors state that they also evaluate the functional impact of these issues, but no data (e.g., Barthel's index, mRS, IADL) is reported, which would suggest what the association is with post-stroke functional status; I suggest that these data be included or the text amended accordingly].

Response 8: 
[Dear Reviewer, thank you for your insightful feedback. Please be advised that the title has been updated accordingly].

Comment 9: 
[In this work (doi: 10.1007/s10072-024-07815-y), the Authors did not observe a correlation between sleep measures (some of which were also used in this study) and motor learning performance; I suggest citing this and other work in the discussion to explore differences from other literature data. For example, their stroke timeframe was subacute/chronic, whereas, here, the Authors assessed earlier stages. What are the potential implications?].

Response 9: 
[Dear Reviewer, Thank you for your insightful feedback. Please be advised that the primary focus of our study is the prevalence of sleep disorders in stroke survivors, which serves as an initial step toward addressing this important topic in our country. As such, we did not comprehensively address the association between sleep disorders and motor skills in the discussion].

Comment 10:
[Amongst the limitations, I also suggest reporting that all measures used are subjective, while there are no objective data (e.g., polysomnography, actigraphs) to confirm the actual presence of sleep disturbances].

Response 10: 
[Dear Reviewer, thank you for your insightful feedback. Please find the added comments on page 14, lines 484-486].

Round 2

Reviewer 3 Report

Comments and Suggestions for Authors

I thank the Authors for their work. No further comments

Author Response

Thank you.